# ASyMOB: Algebraic Symbolic Mathematical Operations Benchmark

**Michael Shalyt** [* 1]  **Rotem Elimelech** [* 1]  **Ido Kaminer** [1]

## Abstract

Large language models (LLMs) are increasingly applied to symbolic mathematics, yet existing evaluations often conflate pattern memorization with genuine reasoning. To address this gap, we present **ASyMOB**, a high-resolution dataset of *35,368* validated symbolic math problems spanning integration, limits, differential equations, series, and hypergeometrics. Unlike prior benchmarks, **ASyMOB** systematically perturbs each seed problem using symbolic, numeric, and equivalence-preserving transformations, enabling a fine-grained assessment of generalization. Our evaluation reveals three key findings: (1) most models' performance collapses under minor perturbations, while top systems exhibit an apparent *regime shift* in robustness; (2) integrated code tools stabilize performance, particularly for weaker models; and (3) we identify examples where Computer Algebra Systems (CAS) fail while LLMs succeed, as well as problems solved only via a hybrid LLM-CAS approach, highlighting a promising integration frontier. **ASyMOB** serves as a principled diagnostic tool for measuring and accelerating progress toward building verifiable, trustworthy AI for scientific discovery.

## 1. Introduction

In recent years, large language models (LLMs) have shown remarkable capabilities in domains such as mathematical reasoning (Lewkowycz et al., 2022; Kojima et al., 2022; Wang et al., 2023; Trinh et al., 2024; Luo et al., 2025; Davies et al., 2021) and code generation (Rozière et al., 2024; Ridnik et al., 2024; Zan et al., 2023; Hou et al., 2024). A crucial skill for real-world applications of these capabilities is mastery of university-level symbolic mathematics, including integration, limit computation, differential equation solving, and algebraic simplification. This proficiency is fundamental across many mathematical, scientific, and engineering challenges.

However, existing mathematical benchmarks inadequately assess symbolic proficiency. Early benchmarks like GSM8K (Cobbe et al., 2021) and MATH (Hendrycks et al., 2021), while driving progress in arithmetic reasoning, focus on pre-university-level questions and have been saturated by frontier LLMs (Glazer et al., 2024). Many popular benchmarks rely on multiple-choice questions (Rein et al., 2024), an unrealistic setting which artificially lowers the difficulty. Word-problem benchmarks mix two fundamentally different challenges: text-to-math conversion (building expressions from text) and symbolic manipulation (solving them). This conflation makes it hard to evaluate an LLM's performance specifically on the latter, and to diagnose the root causes of model errors. Conversely, formal proof datasets (Zheng et al., 2022; Balunović et al., 2025) address theorem proving but often skip core tasks like integration or solving differential equations.

The broad topic coverage that most benchmarks strive for forces small sample sizes per skill category, hindering robust statistical analysis. E.g., in MathBench (Liu et al., 2024) only 4% of the questions address university-level math in English. The 5K test dataset by (Lample & Charton, 2020) targets symbolic math, but mainly contains basic problems. Recent efforts, such as FrontierMath (Glazer et al., 2024) and Humanity's Last Exam (Center for AI Safety et al., 2026), demand that LLMs exhibit very high proficiency across numerous skills simultaneously, thereby impeding conclusions regarding specific LLM capabilities. Overcoming these limitations can shed light on a fundamental question: do LLMs solve problems through genuine mathematical understanding or merely through advanced pattern recognition (Mirzadeh et al., 2025; Boye & Moell, 2025; Zhou et al., 2024b; Huang et al., 2025; Zhou et al., 2025; Jiang et al., 2024). Addressing this question calls for different types of datasets, which can separate sophisticated pattern memorization from mathematical abilities.

In response, we present ASyMOB: Algebraic Symbolic Mathematical Operations Benchmark (pronounced Asimov, in tribute to the renowned author). ASyMOB assesses LLM capabilities through systematic perturbations of core

Project repository:
https://github.com/RamanujanMachine/ASyMOB

[*]Equal contribution  [1]Technion - Israel Institute of Technology, Haifa 3200003, Israel. Correspondence to: Michael Shalyt <shalyt@technion.ac.il>.

*Proceedings of the 43rd International Conference on Machine Learning*, Seoul, South Korea. PMLR 306, 2026. Copyright 2026 by the author(s).

symbolic tasks, introducing three key innovations:

1. **Focused Scope**: Targeting pure symbolic manipulation (Figure 1).

2. **Controlled Complexity**: Systematically introduced questions varied by gradual difficulty levels.

3. **High Resolution**: The large scale and fine-grained difficulty steps enable statistically robust measurement of model accuracy, sensitivity to noise types, and impact of tool use.

---

**Seed Question**

<Code / No-Code Prompt>

*Solve the following integral.*

$$\int_1^2 \frac{e^x(x-1)}{x(x+e^x)}dx$$

**Solution:**

$$\ln\left(\frac{2+e^2}{2+2e}\right)$$

---

**Symbolic Perturbation**

<Code / No-Code Prompt>

*Solve the following integral.*
*Assume A, B, F, G are real and positive.*

$$\int_1^2 \frac{Ae^{Fx}(Fx-1)}{Fx\left(Be^{Fx}+FGx\right)}dx$$

**Solution:**

$$\frac{A}{BF}\cdot\ln\left(\frac{e^2B+2G}{2(eB+G)}\right)$$

---

| No-Code Prompt | *Assume you don't have access to a computer, and do not use code to solve the question.* |
|---|---|
| Code Prompt | *Please use Python to solve the question.* |

*Figure 1.* **Example seed question (top), its symbolically perturbed variant (middle), and code-use preambles (bottom).** The preamble either disallows or encourages code execution (omitted for models without inherent code execution capabilities).

While there are examples of variational math problem generation (Mirzadeh et al., 2025; Li et al., 2024), ASyMOB introduces three key contributions. First, it evaluates university-level symbolic mathematics - whereas other works remain confined to school-level math, mostly derived from GSM8K and MATH. Second, it introduces previously unexplored perturbation categories: 'Symbolic' and 'Equivalence' perturbations probe distinct robustness

dimensions absent in prior work. Third, it focuses on mathematical reasoning rather than linguistic variation, in contrast to GSM-Symbolic (Mirzadeh et al., 2025), whose perturbations primarily alter and measure textual phrasing.

Using ASyMOB, we evaluated the performance of open- and closed-weight LLMs, including general and mathematical models. Perturbations significantly challenge LLMs' symbolic math skills, reducing the average model success rate from 74.6% on the unperturbed subset to 46.8% on the full ASyMOB benchmark. Even the simplest perturbations noticeably affect performance (Figure 2).

Following the extensive investigation of tool use in math problem solving (Novikov et al., 2025; Yue et al., 2024; Zhou et al., 2024a; OpenAI, 2025c; Liao et al., 2024; Gou et al., 2024; Imani et al., 2023; Romera-Paredes et al., 2023; Dugan et al., 2024; Gao et al., 2023; Zhang et al., 2025), we tested code-integrated LLMs both with and without code execution (Figure 2 left) - measuring the effect of tool use in the new context of purely symbolic math challenges. Tool use boosts performance in weaker models, but surprisingly has no effect on advanced ones.

Some perturbed variants in ASyMOB proved impossible for the CAS we tested - Mathematica, WolframAlpha and SymPy (Wolfram Research Inc., 2024; Wolfram Alpha LLC, 2025; Meurer et al., 2017) - yet certain LLMs managed to solve them (section 3.1). Moreover, we present an example where pure CAS and pure LLM approaches fail, yet their combination successfully solves the challenge, leveraging the complementary strengths of each system.

## 2. Methodology for Symbolic Mathematical Operations Measurement

### 2.1. Dataset Design and Generation

We begin by curating and creating a set of 100 seed problems that contain only symbolic content - no word-problems or other textual or graphical information beyond instructions and assumptions. This restriction distinguishes ASyMOB from existing benchmarks (where purely symbolic questions are rare) and most math datasets. 55 seed questions were curated from university-level benchmarks (Chernyshev et al., 2025; Fang et al., 2024; Frieder et al., 2023; Xu et al., 2025) and math olympiads (Brazilian Mathematical Olympiad, 2019; Huang et al., 2024; He et al., 2024). 45 additional seed questions were created to cover underrepresented categories. The questions represent a sample of the practical mathematical challenges that engineers and scientists frequently encounter in their work and research. Each question is categorized by its topic: Integrals (30), Differential Equations (23), Series (22), Limits (15), Hypergeometric Functions (10). Based on these seed questions, we introduce systematic pertur-

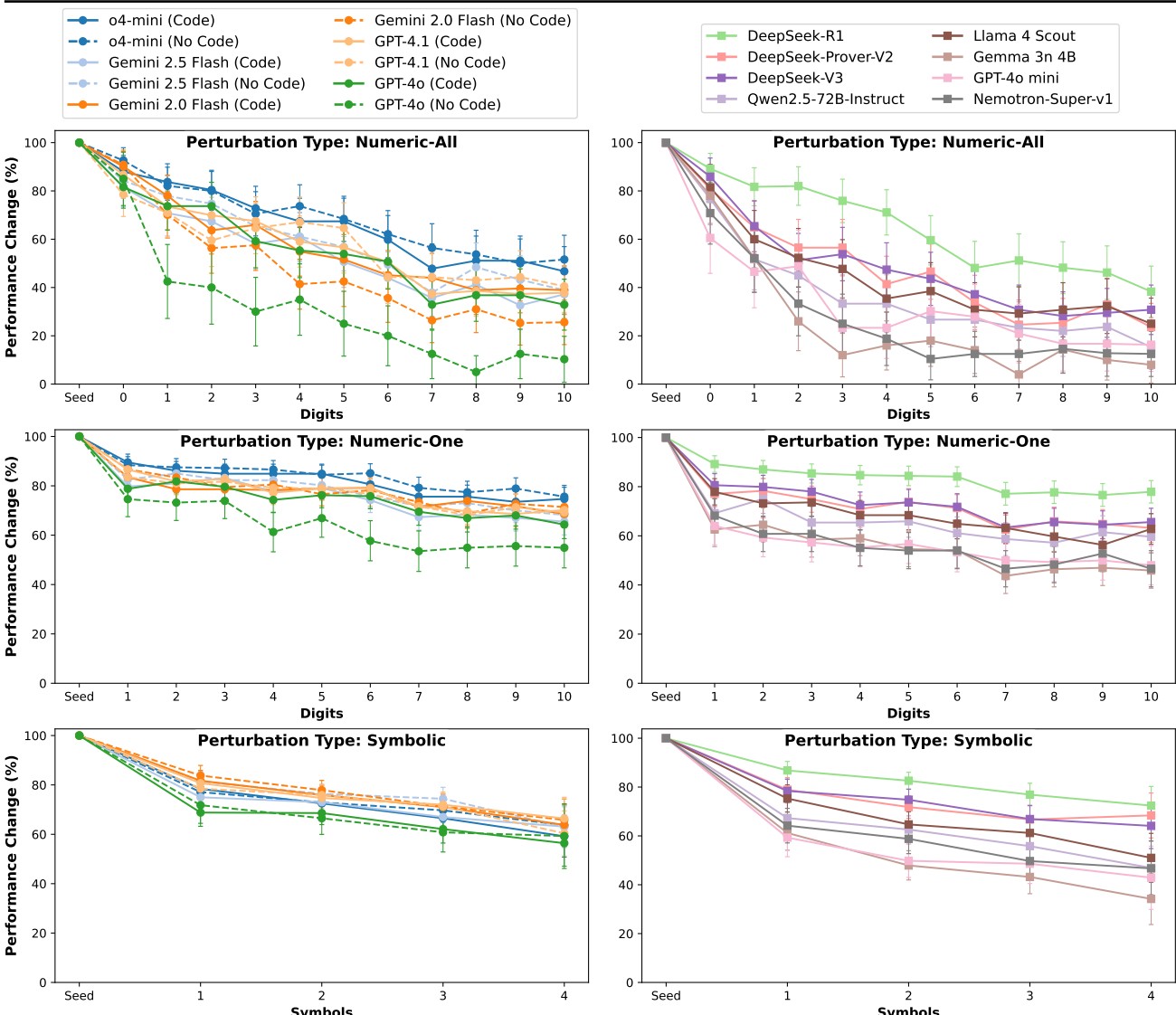

*Figure 2.* **Degradation of success rate relative to seed-set performance.** Both code-integrated models (left) and non-code integrated (right) exhibit performance degradation due to 'Numeric' and 'Symbolic' perturbations, but frontier models are more resilient. Notably, GPT-4o is substantially more robust when code-enabled. Wald 95% confidence intervals are shown (Wald, 1943).

bations to create an overall dataset of 35,368 unique symbolic math challenges (Table 1). The guiding principle was to modify the symbolic structure of the problem - thereby adding a layer of variation - *without substantially altering the core mathematical challenge* or the required solution techniques.

For instance, consider the elementary integral $\int x^2 e^x dx = e^x \left( x^2 - 2x + 2 \right)$, typically solved via integration by parts. An acceptable perturbation is $\int x^2 e^{Fx} dx = \frac{e^{Fx}\left( F^2 x^2 - 2Fx + 2 \right)}{F^3}$. Although this variant introduces a substitution step ($t = Fx$), the fundamental solution technique is preserved. Conversely, a modification like $\int x^{2B} e^x dx = (-x)^{-2B} x^{2B} \Gamma(2B + 1, -x)$ would *not* be considered a symbolic *perturbation* as it significantly increases the problem's complexity and demands additional solution skills.

After manually perturbing each seed question with 2-to-5 symbolic parameters, additional variants were generated using algorithmic transformations. Note that the random nature of the question generation methods makes ASyMOB inherently resilient against benchmark hacking, data leakage, and memorization. The dataset can (and should) be re-generated before assessing a new LLM - unlike static benchmarks which quickly become contaminated.

One of the questions we aim to investigate is the effect of the number of symbolic perturbations on model performance. Specifically, we ask whether each additional perturbation further degrades performance, or whether most of the added difficulty for LLMs arises from the introduction of the first symbolic perturbation - transforming the problem to contain non-numeric parameters. To enable this measurement, we systematically remove added

*Table 1.* **ASyMOB question variants.** For each variant type, the rightmost column shows the number of variants for this seed question and the dataset total (e.g. 30 'Numeric-One-N' variants of this seed, 3490 overall). XX/YY/ZZ in 'Numeric-All-N-S' are 2-digit random numbers. Full dataset available in the supplementary material.

| Variant | Example Challenge | Answer | # |
|---|---|---|---|
| **Seed (Original)** | $\lim_{x\to 0}\left(\frac{2\cdot\tan\left(\frac{x}{2}\right)}{x}\right)^{\frac{3}{x^2}}$ | $e^{\frac{1}{4}}$ | 1 (100) |
| **Symbolic-N** (Shown for N=3) | $\lim_{x\to 0} A \cdot \left(\frac{2\cdot\tan\left(\frac{B\cdot x}{2}\right)}{B\cdot x}\right)^{\frac{C\cdot 3}{(B\cdot x)^2}}$ | $A \cdot e^{\frac{C}{4}}$ | 7 (1348) |
| **Numeric-All-N** (Shown for N=2) | $\lim_{x\to 0} 17 \cdot \left(\frac{2\cdot\tan\left(\frac{91\cdot x}{2}\right)}{91\cdot x}\right)^{\frac{57\cdot 3}{(91\cdot x)^2}}$ | $17 \cdot e^{\frac{57}{4}}$ | 11 (1100) |
| **Numeric-One-N** (Shown for N=6) | $\lim_{x\to 0}\left(\frac{2\cdot\tan\left(\frac{x}{2}\right)}{x}\right)^{\frac{838310\cdot 3}{x^2}}$ | $e^{\frac{838310}{4}}$ | 30 (3490) |
| **Numeric-All-N-S** (Shown for N=2) | $\lim_{x\to 0} \text{XX} \cdot \left(\frac{2\cdot\tan\left(\frac{\text{YY}\cdot x}{2}\right)}{\text{YY}\cdot x}\right)^{\frac{\text{ZZ}\cdot 3}{(\text{YY}\cdot x)^2}}$ | $\text{XX} \cdot e^{\frac{\text{ZZ}}{4}}$ | 100 (10000) |
| **Equivalence-One-Easy** | $\lim_{x\to 0+}\left(\frac{2\cdot\tan\left(\frac{x}{2}\right)}{x}\right)^{\frac{(\sin^2(-Fx)+\cos^2(Fx))\cdot 3}{x^2}}$ | $e^{\frac{1}{4}}$ | 15 (1745) |
| **Equivalence-One-Hard** | $\lim_{x\to 0+}\left(\frac{\sinh\left(\log\left(Ax+\sqrt{A^2x^2+1}\right)\right)}{Ax}\right)\left(\frac{2\cdot\tan\left(\frac{x}{2}\right)}{x}\right)^{\frac{3}{x^2}}$ | $e^{\frac{1}{4}}$ | 15 (1745) |
| **Equivalence-All-Easy** | $\lim_{x\to 0+}(\sin^2(-Ax)+\cos^2(Ax))\left(\frac{2\cdot\tan\left(\frac{(-\sinh^2(Bx)+\cosh^2(Bx))x}{2}\right)}{(-\sinh^2(Bx)+\cosh^2(Bx))x}\right)^{\frac{\left(\frac{\ln(x)\cdot\log_x(F)}{\ln(F)}\right)3}{((-\sinh^2(Bx)+\cosh^2(Bx))x)^2}}$ | $e^{\frac{1}{4}}$ | 60 (7920) |
| **Equivalence-All-Hard** | $\lim_{x\to 0+}\left(\frac{\tan(x)+\tan(x(A-1))}{(-\tan(x)\tan(x(A-1))+1)\tan(Ax)}\right)\left(\frac{2\cdot\tan\left(\left(\frac{\frac{\log_B\left(\frac{x}{x}\right)+\log_B(e)}{\log_B(e)}\right)x}{2}\right)}{\left(\frac{\log_B\left(\frac{x}{x}\right)+\log_B(e)}{\log_B(e)}\right)x}\right)^{\frac{\left(\frac{x\sum_{\cdots}\frac{x}{x\cdot k+1}}{\cdots}\right)3}{\left(\left(\frac{\log_B\left(\frac{x}{x}\right)+\log_B(e)}{\log_B(e)}\right)x\right)^2}}$ | $e^{\frac{1}{4}}$ | 60 (7920) |

symbols from each manually perturbed question, generating all possible combinations. This approach helps avoid subjective bias in perturbation choice. Each variant is labeled as 'Symbolic-N', where $N$ indicates the number of perturbing symbols. For example, a question originally marked as 'Symbolic-4' will yield additional variants: four 'Symbolic-3', six 'Symbolic-2', and four 'Symbolic-1'.

Mathematically, if a model can solve a symbolically perturbed question, it should also be able to solve its numeric counterpart by substituting constants with symbols, solving symbolically, and substituting back. Yet, as Figure 2 shows, LLMs often underperform on numeric perturbations compared to symbolic ones, suggesting their reasoning remains constrained by their token-based architectures.

To test this, numeric variants were created by replacing every symbolic parameter with a random positive integer of fixed digit length, varying from 0 to 10 digits to probe both in- and out-of-distribution performance (very large coefficients are rare in training). Here, 0 digits means replacing all symbols by '1', yielding a mathematically equivalent question - yet, Figure 2 shows even this trivial case degrades performance, further questioning LLMs' true mathematical understanding. These variants are labeled 'Numeric-All-N', where $N$ is the digit length.

Due to the probabilistic nature of LLMs, we measure stability over 50 random variations of 'Numeric-All-N' for $N = 2, 3$ - generating a new set of random 2 or 3-digit numbers per variation (Figure 9 in Appendix C). These variants are marked as 'Numeric-All-N-S'.

To explore whether the initial introduction of a large number causes a disproportionate performance drop, or whether performance declines progressively with each added numeric coefficient, we also create variants (labeled 'Numeric-One-N') where only one symbolic parameter is replaced by a number (ranging from 1 to 10 digits), and the remaining symbols are removed. To avoid bias, we generate all possible choices of which symbol to replace.

Numeric perturbations are similar in spirit to previous works like (Mirzadeh et al., 2025; Zhang, 2026; Shrestha et al., 2025; Srivastava et al., 2024; Huang et al., 2025) - which are based on GSM8K (Cobbe et al., 2021) or MATH (Hendrycks et al., 2021) word problems, as well as (Balunović et al., 2025) - that focuses on constructive proofs. Differing from these previous benchmarks, the larger-scale ASyMOB dataset focuses on advanced symbolic math problems, with no language understanding component, and applies controlled added complexity.

Finally, we evaluate the impact of equivalent-form pertur-

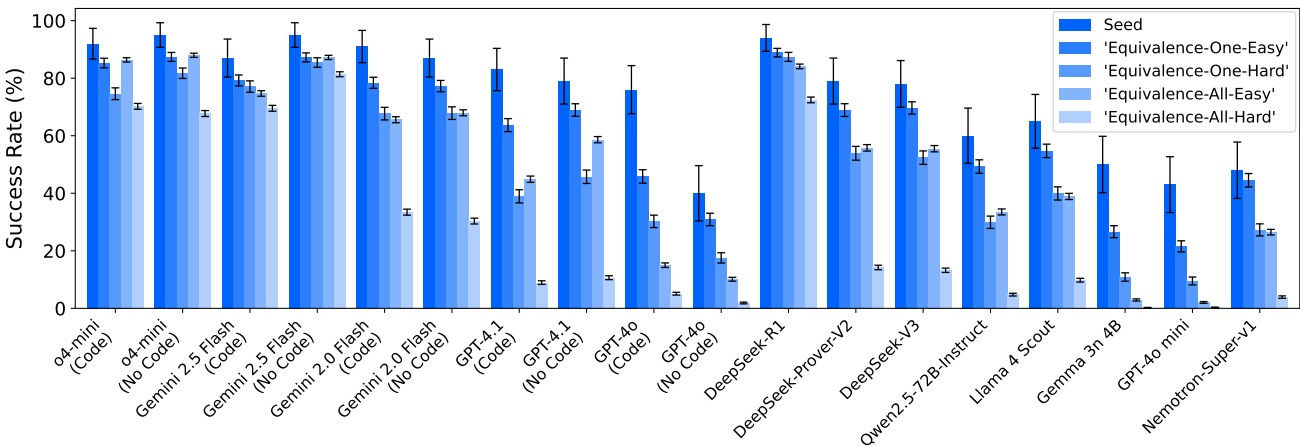

*Figure 3.* **Effect of 'Equivalence' perturbations.** Note the substantial drop in success rate vs. seed set performance for most models. Wald 95% confidence intervals are shown (Wald, 1943).

bations. In this case, we complicate the problem by inserting one or more expressions that are mathematically equal to 1, and are publicly known (available on Wikipedia). For example, symbol $A$ might be replaced by $\sin^2(-Ax) + \cos^2(Ax)$. While such perturbations (intentionally) introduce extra steps in simplification, the final answer is identical to the seed version. We confirmed that the tested LLMs could correctly simplify each expression when presented individually, indicating that the 'Equivalence' variants' additional difficulty arises from the increased structural complexity - not the added expressions themselves.

Five identity types were selected for this transformation - trigonometric, hyperbolic, logarithmic, complex exponential, and series - each with an 'Easy' and a 'Hard' version (see Appendix A.1 and Table 3). The 'Easy'/'Hard' classification was defined according to the intrinsic complexity of the identity, including the number of nested operations and the algebraic simplification steps required (the performance drops in Figure 3 validate this classification). To generate equivalence variants, these identities replace the symbols in the symbolic perturbations. To bound the effect of such perturbations, while limiting the combinatorial explosion of question variants, we choose two extremes: either all symbols are replaced ('Equivalence-All-Easy/Hard'), or only one symbol is replaced ('Equivalence-One-Easy/Hard').

Some resulting variants may appear atypical of how problems are usually posed. This complexity is intentional. The 'Hard' Equivalence perturbations are specifically designed to induce expression swell, in addition to higher baseline difficulty. While such formulations are rarely posed directly in exams (or other benchmarks), similar sub-expressions routinely arise as intermediate steps in extended derivations, for example after substitutions, algebraic rewrites, or integration by parts. These cases require the ability to simplify and restructure complex symbolic components before attempting to solve the full problem.

We view this capability to *think in steps* as a core aspect of mathematical (and general) reasoning. Therefore these stress-test perturbations are treated as diagnostic probes of an LLM's capacity to reason, manage long-range algebraic dependencies, and perform symbolic simplification, rather than as direct proxies for problem distributions.

Figure 3 shows that for most LLMs the challenge level of a single 'Hard' perturbation is lower than multiple 'Easy' perturbations - but not for all LLMs. The underlying causes of this difference remain a topic for future investigation.

One of the advantages of ASyMOB is once the seed and manual symbolic perturbations are complete and thoroughly validated, all other tasks are generated algorithmically - removing the risk of human errors in specific questions or answers. This is not obvious, as existing mathematical benchmarks are known to have up to 5-10% mistaken labeling and formatting errors (Vendrow et al., 2024; Zhang et al., 2026; Patel et al., 2021). See Appendix A.2 for examples discovered during the ASyMOB seed curation process. Additionally, by maintaining consistent question formatting and disallowing substantial textual or graphical information, we prevent potential task ambiguities and missing data (Vendrow et al., 2024).

## 2.2. Testing and Validation

Validating open-ended symbolic problems is harder than closed-form or numerical ones. E.g., the reference answer to question #51 in the ASyMOB dataset is $\frac{1}{2}\sqrt{x}$. However, solving it using Mathematica yields $e^{\frac{1}{2}(\log(x)-2\log(2))}$. Although structurally different, these expressions are mathematically identical. Our evaluation must accept any correct symbolic form and phrasing without penalizing the LLM (e.g. '$\sqrt{x} \cdot \frac{1}{2}$', '$y = \frac{1}{2}\sqrt{x}$', '$y \to \frac{1}{2}\sqrt{x}$', etc.). To prevent false negatives, we implement a multi-step validation process with dual verification methods (Figure 4).

The final mathematical answer is extracted from the LLM's

*Table 2.* **Model performance on ASyMOB by perturbation category**. Bold indicates the top performer in each category. Subset titles are color-coded in accordance to Table 1. The bottom line shows SymPy success statistics, providing pure CAS performance baseline. Note that SymPy's 100% success rate on the Seed set is unsurprising, as we validated all questions during seed selection and creation via CAS, introducing a natural selection bias in favor of SymPy.

| Model | Seed | Symbolic | Numeric | Equivalence | Variance | Total |
|---|---|---|---|---|---|---|
| **Closed-Weights Models** | | | | | | |
| o4-mini (code) | 92 | 69.0 | 74.9 | 78.6 | 72.8 | 76.1 |
| o4-mini (no code) | **95** | 71.8 | **78.1** | 79.0 | **76.8** | 78.1 |
| GPT-4.1 (code) | 83 | 66.1 | 66.3 | 31.3 | 62.8 | 46.2 |
| GPT-4.1 (no code) | 79 | 64.7 | 64.8 | 38.7 | 58.8 | 48.9 |
| GPT-4o (code) | 76 | 57.1 | 61.3 | 15.1 | 59.3 | 35.3 |
| GPT-4o (no code) | 40 | 34.5 | 32.3 | 9.3 | 21.6 | 16.8 |
| GPT-4o-mini | 43 | 26.9 | 27.6 | 3.8 | 17.6 | 11.8 |
| Gemini-2.5 Flash (code) | 87 | 70.3 | 68.2 | 73.2 | 62.6 | 69.5 |
| Gemini-2.5 Flash (no code) | **95** | 75.9 | 72.6 | **84.7** | 69.5 | **78.5** |
| Gemini-2.0 Flash (code) | 91 | 71.9 | 68.2 | 53.7 | 59.7 | 58.1 |
| Gemini-2.0 Flash (no code) | 87 | 69.7 | 64.1 | 53.4 | 51.2 | 54.9 |
| **Open-Weights Models** | | | | | | |
| DeepSeek-V3 | 78 | 64.2 | 59.5 | 39.2 | 48.2 | 45.4 |
| DeepSeek-R1 | 94 | **78.8** | 76.7 | 80.1 | 75.2 | 78.3 |
| DeepSeek-Prover-V2-671B | 79 | 65.6 | 59.8 | 39.8 | 50.1 | 46.3 |
| Llama-4-Scout-17B | 65 | 50.6 | 48.2 | 28.5 | 36.7 | 34.3 |
| Qwen2.5-72B-Instruct | 60 | 45.3 | 43.5 | 22.8 | 29.1 | 28.2 |
| Gemma-3n-e4b-it | 50 | 30.4 | 30.3 | 4.7 | 15.1 | 12.0 |
| Nemotron-Super-49B-v1 | 48 | 37.1 | 34.0 | 18.9 | 23.6 | 23.0 |
| SymPy | 100 | 56.7 | 65.2 | 21.9 | 57.8 | 39.2 |

*Figure 4.* **Result validation process.** Extract LaTeX answer via regex; parse to SymPy (deterministic, fallback to LLM); validate symbolically and numerically against reference answer.

full textual response using a highly flexible regular expression (Appendix B). The extracted LaTeX expression is then cleaned (e.g. formatting commands like \displaystyle and \boxed are removed) and parsed into a SymPy expression using `sympy.parsing.latex.parse_latex`. If the parsing fails, we resort to using gemini-2.0-flash (Pichai et al., 2024) for this translation (occurred in ∼9% of cases). Note that this fallback is used only to translate extracted LaTeX answers into SymPy objects - it is never used to judge mathematical correctness, and the LLM never sees the full answer, only the final LaTeX expression. Since final answers are simpler expressions than the problems themselves, LaTeX⇒SymPy translation is easier than the original challenge, and relies on the model's coding skills, not mathematical prowess - so a model can translate into code even answers to problems which it cannot solve.

The resulting SymPy expression undergoes two distinct validation checks against the reference answer (also represented as a SymPy object):

- **Symbolic validation**. The difference between the extracted expression and the correct answer is simplified via `SymPy.simplify`. If the simplification reduces this difference to zero (or a constant, in the case of in-

definite integrals), the answer is deemed correct.

- **Numeric validation**. We randomly generate numerical values for each variable (e.g., $x$ and any symbolic perturbation parameters) and substitute them into both the LLM's expression and the reference answer. If the relative difference between the two evaluations is less than 0.002%, the answers are considered matching. This process is repeated 5 times to mitigate the risk of coincidental matches. To allow the detection of numeric equivalence between indefinite integrals, we require that all 5 repetitions produce the same difference (not necessarily zero), concluding that the expressions are equivalent up to a constant factor.

This validation strategy avoids employing LLMs as judges - as was done in (Chernyshev et al., 2025) and (Fang et al., 2024) - thus avoiding validation errors due to LLM pattern recognition biases, as was shown to happen (Mao et al., 2024; Chernyshev et al., 2025).

We exclusively use the pass@1 evaluation criterion, reflecting the practical requirement for reliability in real-world applications by engineers and researchers. The inherent LLM randomness is accounted for by evaluation across the large number of questions within each category.

## 3. Experimental Results

Using ASyMOB, open/closed-weight, general and math-specialized LLMs were evaluated (Table 2). While advanced closed-weight models - o4-mini, Gemini 2.5 Flash (OpenAI, 2025b; Kavukcuoglu, 2025) - achieve the highest seed accuracy, older models - Gemini 2.0 Flash, GPT-4.1, GPT-4o, GPT-4o-mini (Pichai et al., 2024; OpenAI, 2025a; 2024) - and open-weight models - DeepSeek-V3, DeepSeek-R1, DeepSeek-Prover-V2-671B, Llama-4-Scout-17B-16E-Instruct, Qwen2.5-72B-Instruct, Gemma-3n-e4b-it, Llama-3_3-Nemotron-Super-49B-v1 (DeepSeek-AI, 2025b;a; Ren et al., 2025; Meta, 2025; Yang et al., 2024; Farabet & Warkentin, 2025; Bercovich et al., 2025) - also perform reasonably well.

A significant finding is the substantial degradation in performance when models are faced with perturbed versions of the seed questions (Figures 2, 3). Some LLMs struggle more with symbolic perturbations, while others falter with numeric perturbations. Understanding the reasons behind these differences between models may reveal deeper principles of how LLMs process mathematical structures.

A possible cause for the performance degradation is the increased length of input token chains for perturbed variants vs. seed problems - but on its own it does not explain the full degradation. For example, 'Numeric-All-0' variants add just a few characters, increasing the length of some expressions by only 3–4%, yet producing 50–100% relative degradation in some cases. Conversely, we observe cases where substantially larger character increases do not degrade performance. Comparing 'Symbolic' and 'Numeric-1' perturbations, which add the same number of characters, isolates the effect of perturbation type from length. We observe a distinct performance gap (symbolic perturbations are harder), providing further evidence that models struggle with the mathematical substitutions, and not merely with token count.

The top models excel in robustness to perturbations - arguably a more critical measure of LLM generalization - achieving a 20% performance gap between o4-mini,

Gemini-2.5 Flash, and DeepSeek-R1, and the next-best model on the full dataset. This robustness holds across seeds, perturbation categories, and mathematical topics, including out-of-distribution settings (Figure 5, see Appendix C for further discussion). Such consistency can suggest a regime shift in robustness, with newer models exhibiting resilience beyond what their baseline performance would predict (Figure 6).

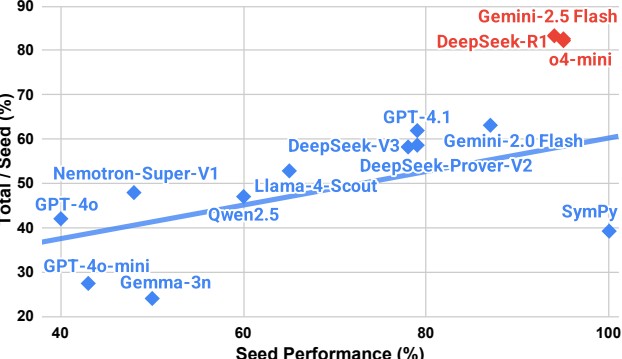

*Figure 6.* **Do models exhibit a robustness regime shift?** Perturbation resilience (performance on the total dataset divided by seed performance) vs. seed performance, without tool use. Resilience increases with baseline performance across all models; however, the three strongest models exhibit a disproportionately larger gain relative to the trend observed for the remaining models.

Comparing LLM performance on ASyMOB to recent competitive math benchmarks, like AIME2025 (Balunovic et al., 2025) and RIMO (Chen et al., 2025), we see a general correlation. However, we notice that DeepSeek-Prover-V2-671B - despite achieving 88.9% pass ratio on the MiniF2F proof benchmark (Zheng et al., 2022; Ren et al., 2025) and outperforming both Gemini-2.5 and DeepSeek-V3 on PutnamBench (Tsoukalas et al., 2024) - is still surpassed by DeepSeek-R1 (from the same model family), on every category in ASyMOB. Furthermore, its performance gains vs. the base model (DeepSeek-V3) are incremental at best. This suggests that proficiency in proof generation may not directly translate to skill in symbolic mathematical operations, where the model's reasoning capabilities can prove more effective. Nemotron-Super (Bercovich et al., 2025), on the other hand, shows relatively high perturbation re-

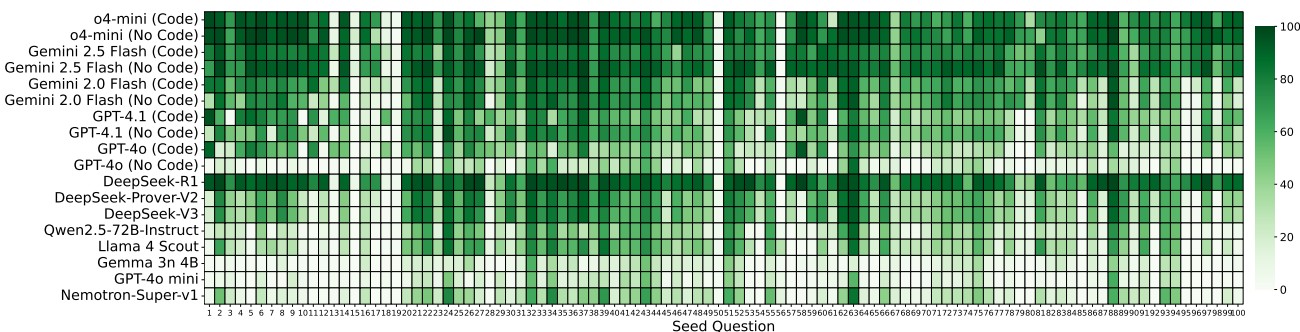

*Figure 5.* **Heatmap of overall performance per model per seed, averaged over all perturbations.**

silence, despite the low success rate on the seed subset.

The 'Variance' subset provides insights into model consistency. The variance of results over all 'Numeric-All-N-S' variants was calculated per seed question and per model (Figure 9). An interesting observation is the absence of correlations in variance across models for the same seed question, indicating that the effect of perturbation is similar regardless of the specific seed (see Appendix C.3).

Enabling code execution improved the performance of older models (GPT-4o by up to 37.7% and Gemini-2.0 Flash by up to 8.6% in a single category), likely compensating for their symbolic-math weaknesses through coding skills. In contrast, frontier models performed similarly or worse with code execution, likely because their limitations become apparent on the hardest problems - which are usually unsolvable by a naive application of SymPy - so gains require combining the model's internal reasoning (to break down complex problems) with strategic tool use. Both effects highlight the value of hybrid solution strategies.

### 3.1. Computer Algebra Systems Limitations

While CAS like SymPy, Mathematica, and WolframAlpha are powerful tools for symbolic mathematics, ASyMOB includes instances where traditional CAS fail yet LLMs succeed. E.g., 2 of the 5 'Hard Equivalence' forms (Appendix A.1) are not recognized by SymPy as identical to 1. Yet, many variants containing these identities are solved by models in our testing. Another example is the question in Table 1 - where WolframAlpha does not simply fail to answer on variant 'Symbolic-3', but produces a false result[1]. Such examples showcase the need for LLMs proficient in symbolic mathematics that can overcome CAS limitations.

Figure 7 shows perhaps the most instructive example. Pure CAS and pure LLM approaches both failed. However, when instructed to simplify the integral first and then solve using CAS, the model succeeded, demonstrating the power of combining LLM strategic ability with CAS rigor.

## 4. Discussion and Outlook

We introduced ASyMOB, a high-resolution symbolic mathematics benchmark that isolates core symbolic reasoning skills, containing 35,368 challenges. Assessment of leading models shows:

- LLMs' symbolic math performance substantially degrades under perturbations, suggesting reliance on pattern memorization.

- Top LLMs show a leap in robustness against perturbations of various kinds, suggesting strong symbolic math generalization capabilities.

- Correct tool use (code execution) can meaningfully improve performance, especially when applied via hybrid LLM+CAS strategies.

Benchmarks aspire to present uncontaminated "new" questions, fearing memorization of fixed question-answer pairs from public data, but ASyMOB bypasses this challenge via systematic perturbations. Even if a model was trained on some of the seed questions, the benchmark results remain meaningful - an important property as sourcing truly novel questions becomes increasingly infeasible for large-scale datasets.

To empirically assess this robustness, we ran experiments on Gemini 2.0 Flash, OpenAI GPT-4o, and LLaMA 3.3 Nemotron Super, explicitly including the original seed question and its answer as an in-context exemplar within the prompt. While performance improved on simple perturbations (Numeric-All-0: +2%, +27%, +43.5% respectively), the effect quickly dropped on more complex ones (average over Numeric-One-3, Numeric-All-3, Symbolic-3: +2%, +5.1%, +6.8% respectively). These findings show that seed question contamination does not substantially distort performance on harder variants, and ASyMOB's complex perturbations still expose limitations beyond memorization. Given the extremity of this setup (akin to one-shot fine-tuning *per question*), these modest gains likely represent an upper bound compared to pretraining, underscoring ASyMOB's robustness.

This stress test does not rule out "methodological contamination" - training directly on ASyMOB-style perturbation procedures. But such "contamination" can be reframed as a feature. If LLMs improve on re-generated questions by training on previous benchmark iterations, that signifies deeper mathematical understanding - a desirable capability.

Looking forward, LLMs should be intentionally trained to generalize, both via tool use and through perturbations on the training set. Previous works showed that such synthetic data improves overall performance (Li et al., 2024). Fine-grained perturbations provide a systematic method for generating high-quality synthetic data, offering a valuable resource for fine-tuning future reasoning models.

One of our perturbations is inspired by GSM-Symbolic (Mirzadeh et al., 2025) - which showed that even "trivial" complications in *textual* math questions can substantially reduce success rates (up to 65%). Similarly, in our work, *symbolic* complications also led to substantial performance drops (up to 60.9%). This test generalizes the finding of GSM-Symbolic that "current LLMs are not capable of genuine logical reasoning", now shown in the domain of sym-

---

[1]Tested on Wolfram Language version 14.2.1: `https://www.wolframalpha.com/input?i=Limit%5BA+%28Tan%5B%28B+x%29%2F2%5D%2F%28%28B+x%29%2F2%29%29%5E%28%28C+3%29%2F%28B+x%29%5E2%29%2C+x+-%3E+0%2C+Direction+-%3E+%22FromAbove%22%5D`

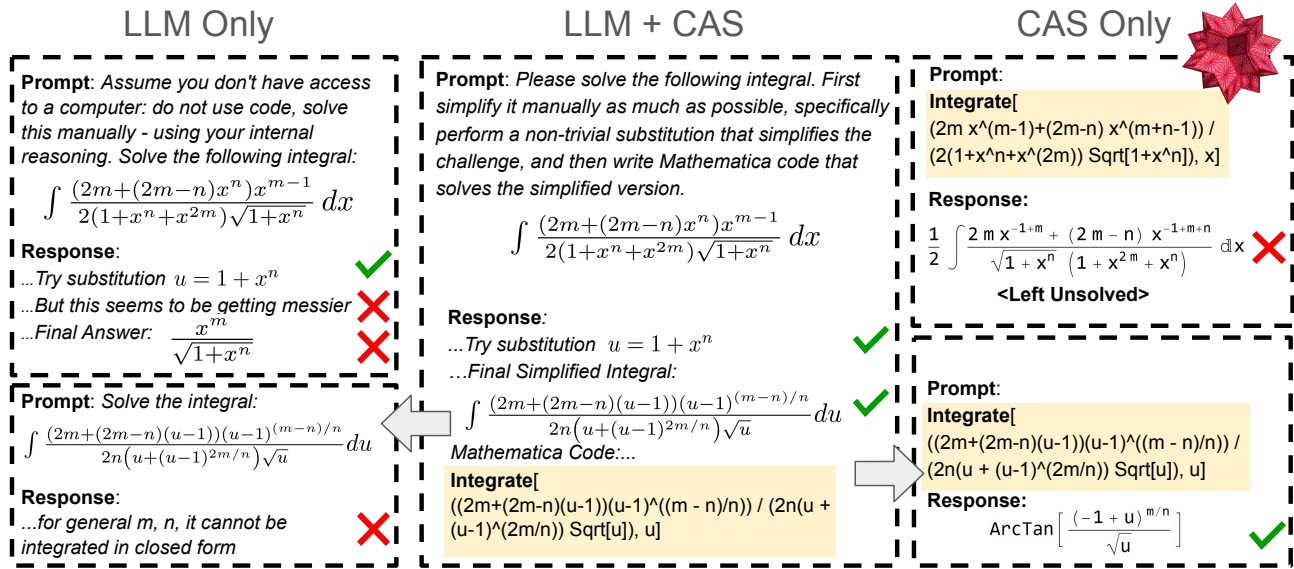

*Figure 7.* **Example question solved exclusively by a hybrid LLM+CAS approach.** ASyMOB's question #122 was solved incorrectly (left) by GPT-4o, despite the model "considering" a correct substitution. Standard CAS systems also failed to solve the question (right). However, a hybrid strategy succeeded: GPT-4o was prompted to first simplify the problem via substitution and then use CAS code on the simplified expression - enabling Mathematica to solve the question.

bolic manipulations and not just in text-to-math conversion.

Importantly, our results suggest a solution hypothesis: once an LLM learns *when* and *where* to use tools, it can potentially mitigate substantial pitfalls by using code execution as a form of grounding. Prompting strategies such as "simplify-then-code" (Figure 7) are one example of this hybrid approach.

Until recently, the hybrid LLM+CAS approach appeared to be the most promising path forward. However, the surprising finding that frontier models *no longer benefit* from CAS use for symbolic math triggers deeper and more fascinating possibilities. Looking ahead, we see three possible trajectories for future developments in AI for math and AI for science:

1. **Intrinsic mastery:** Frontier models may continue to improve in their inherent abilities, eventually surpassing the need for external symbolic math tools, as in the top model behavior observed in this work.

2. **Deeper integration:** Tool use may remain essential, but will demand increasingly sophisticated CAS capabilities that co-evolve with LLMs, complementing their inherent abilities and motivating the next generation of CAS infrastructure.

3. **Autonomous tool creation:** LLMs may internalize symbolic computation itself - leveraging their reasoning and coding capacities to build internal, CAS-like mechanisms that blur the boundary between model and tool.

## Acknowledgments

This research received support through Schmidt Sciences, LLC.

## Impact Statement

As a benchmark for symbolic mathematical reasoning, this work presents limited misuse risk, but can have relevant societal implications. A potential benefit is better calibration of LLM reliability in scientific and engineering workflows, where robust and trustworthy tools could broaden and democratize access to mathematical problem solving and accelerate scientific discovery. A primary risk is automation bias: users may over-trust plausible symbolic outputs from brittle systems. ASyMOB can reduce that risk by exposing perturbation-sensitive failures and encouraging verification-centered evaluation of LLM and LLM-CAS systems.

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

# A. Additional Details About the Dataset

## A.1. List of Equivalence Perturbations

The complete list of 'Equivalence' perturbations, discussed in section 2.1, is provided in Table 3.

| Category | Equivalence Perturbation |
|---|---|
| **Easy** | |
| **Trigonometric** | $\sin^2(-Qx) + \cos^2(Qx)$ |
| **Hyperbolic** | $\cosh^2(Qx) - \sinh^2(Qx)$ |
| **Logarithmic** | $\dfrac{\ln(x)\,\log_x(Q)}{\ln(Q)}$ |
| **Series** | $\dfrac{Q\sum_{N=1}^{\infty}\frac{2^{-N}x}{Q}}{x}$ |
| **Complex Exponential** | $-\dfrac{i(e^{iQx} - e^{-iQx})}{2\sin(Qx)}$ |
| **Hard** | |
| **Trigonometric** | $\dfrac{\tan(x) + \tan(x(Q-1))}{(1 - \tan(x)\tan(x(Q-1)))\tan(Qx)}$ |
| **Hyperbolic** | $\dfrac{\sinh\left(\log(Qx + \sqrt{Q^2x^2+1})\right)}{Qx}$ |
| **Logarithmic** | $\dfrac{\log_Q(x/e) + \log_Q(e)}{\log_Q(x)}$ |
| **Series** | $\dfrac{Q\sum_{N=1}^{\infty}\frac{6x}{\pi^2 N^2 Q}}{x}$ |
| **Complex Exponential** | $-\dfrac{2i(e^{4iQx} + 1)\tan(Qx)}{(1 - e^{4iQx})(1 - \tan^2(Qx))}$ |

*Table 3.* List of the 'Easy' and 'Hard' expressions, which are identical to 1 for any $Q$ and $x$, used in the 'Equivalence' perturbations.

Notably, while the tested LLMs successfully simplified these expressions, SymPy (Meurer et al., 2017) was unable to simplify the more difficult trigonometric and hyperbolic identities to 1, providing another example for CAS limitations in university-level symbolic math challenges.

## A.2. Discovered Benchmark Errors

As mentioned in Section 2.1, existing mathematical benchmarks are known to have up to 5-10% mistaken labeling and formatting errors (Vendrow et al., 2024; Zhang et al., 2026; Patel et al., 2021).

For example, question 97 from the GHOSTS 'Symbolic Integration' subset (Frieder et al., 2023): "What is the integral of $2x - x^7\mathrm{atan}(3)$". The output: "...The antiderivative... $\frac{2x^2}{2} - \frac{1}{7}x^8\mathrm{atan}(3) + C$" receives a 5/5 rating, but the $\frac{1}{7}$ should have been $\frac{1}{8}$, potentially creating false positives.

Another example from OlympiadBench (He et al., 2024), subset 'OE_TO_maths_en_COMP', id:2498: "If $\log_2 x - 2\log_2 y = 2$, determine $y$, as a function of $x$". The dataset provides both a full solution: "...to obtain $y = \frac{1}{2}\sqrt{x}$", and a final answer: "$\frac{1}{2}, \sqrt{x}$". The extra comma that appeared in the middle of the final answer prevents deterministic systems from recognizing correct answers.

We inserted both of these questions (with corrected answers) as two of our seeds.

ASyMOB's algorithmic generation methods substantially reduce the risk for such errors in specific questions or answers.

### A.3. 'Symbolic-N' Subsets Analysis

Due to the requirement that substituting all symbols with 1 reverts the question to its original seed form, the total number of 'Symbolic-N' variations depends on N. For instance, ASyMOB contains only 7 'Symbolic-5' questions. This small sample size is the reason 'Symbolic-5' is not represented in Figure 2, as it is insufficient for robust statistical analysis. This variability also means that the baseline difficulty of 'Symbolic-N' questions changes with different values of N. The 7 seed questions with a maximal perturbation of 5 symbols have an average success rate across all models of 86.6%. In contrast, the 13 seed questions with a maximal perturbation of 4 symbols have a 74.7% success rate, and the overall success rate across all seeds is 73.9%. The 'Symbolic-4' subset includes 13 questions with maximal 'Symbolic' perturbation (derived from the 13 seeds mentioned above) and 35 permutations based on the 7 maximally perturbed 'Symbolic-5' questions. It is likely that the lower initial difficulty of the seeds influences the difficulty of their derived variations to some extent. Therefore, the difficulty of each 'Symbolic' subset should not be assumed to be identical. This effect can account for the slight increase in success rate observed across most models in the bottom graphs of Figure 2 for 3 and 4 symbols.

## B. Testing Details

As noted in section 2.2, a core principle of the test process is to rely on deterministic and predictable tools whenever possible. Figure 4 shows a "Formatting Instructions" wrapper around the challenge text. Specifically, these instructions state:

*"Finish your answer by writing "The final answer is:" and then the answer in LaTeX in a new line. Write the answer as a single expression. Do not split your answer to different terms. Use $$ to wrap the LaTeX text. Do not write anything after the LaTeX answer."*

The primary goal is to encourage the LLM to produce a clear LaTeX expression, labeled with "The final answer is:". We opt against using forced structured outputs, even when available, to ensure a fair comparison with models lacking this capability and to avoid introducing requirements beyond symbolic math skills. In essence, we aim to minimize the impact of specific phrasing and structural choices in both language and mathematical presentation.

Once the full answer is received, a series of regexes are used to extract the final answer:

```
Pattern 1 (as instructed):
    r'\**[Tt]he final answer is:?\**\s*'
    r'(?:(?:\\\()|(?:\\\[)|(?:\$+))'
    r'(.*?)'
    r'(?:(?:\\\))|(?:\\\])|(?:\$+))'

Pattern 2 (last boxed expression):
    r'\\boxed\{(.*?)\}' + '(?:\n|$|")'

Pattern 3 (last display expression):
    r"\$+(.*?)\$+"

Pattern 4 (output=' ' case):
    r"output='(.*?)'"

Pattern 5 (output=" " case):
    r'output="(.*?)"'
```

While the first pattern represents the given formatting instructions - other output formats were accepted as well. It's important to note that responses claiming, for example, the challenge is impossible or asking for specific values to substitute into the symbols, will frequently lack fitting LaTeX expressions. Therefore, the absence of relevant LaTeX usually indicates a missing or incoherent answer, not a parsing issue. Overall, this stage was successful in 98% of cases.

The extracted LaTeX expression is then cleaned and parsed into a SymPy expression using `sympy.parsing.latex.parse_latex`. If the parsing fails, we resort to using an LLM (gemini-2.0-flash) for this translation. It's important to note that not all "final answer" expressions extracted by our permissive regexes are valid LaTeX or even mathematical expressions. Therefore, a failure to produce a working SymPy expression usually indicates a broken or irrelevant answer, rather than a translation issue. Overall, this stage was successful in 96.1% of cases. Responses that could not be translated into valid SymPy expressions after both parsing attempts were excluded from aggregate statistics.

As detailed in Section 2.2, the resulting SymPy expression undergoes two distinct validation checks against the reference answer (also represented as a SymPy object). Due to the limitations of SymPy (imperfections in `SymPy.simplify`, handling of very large numbers in `.evalf()`, etc.), if either validation method confirms an answer, it is treated as correct (false positives are highly unlikely). Out of all the valid SymPy expressions created in the previous stage, 97.6% were successfully tested. Responses that could not be verified by either method due to SymPy's technical limitations were excluded from the data analysis and omitted from the reported statistics.

In terms of resources required for this work, by far the largest cost was querying the LLMs. The specific costs per setup (model with and without code execution) are summarized in Table 4. Dataset generation compute was negligible (less than 5 minutes on a single workstation), while the validation stage was more resource-intensive (∼10 hours on 3 workstations). Note that the validation process is trivially parallelizable.

| Model Name | Cost |
|---|---|
| **Closed-Weights Models** | |
| Gemini-2.0-flash (no code) | $33 |
| Gemini-2.0-flash (code) | $32 |
| Gemini-2.5-flash (no code) | $687 |
| Gemini-2.5-flash (code) | $648 |
| GPT-4.1 (no code) | $524 |
| GPT-4.1 (code) | $1574 |
| GPT-4o (no code) | $545 |
| GPT-4o (code) | $1595 |
| GPT-4o-mini | $16 |
| o4-mini (no code) | $799 |
| o4-mini (code) | $1849 |
| **Open-Weights Models** | |
| Gemma-3n-e4b-it | $22 |
| Llama-4-Scout-17B-16E-Instruct | $273 |
| Nemotron-Super-49B-v1 | $250 |
| Qwen2.5-72B-Instruct | $261 |
| DeepSeek-Prover-V2-671B | $244 |
| DeepSeek-R1 | $927 |
| DeepSeek-V3 | $244 |

*Table 4.* Cost of evaluating each test setup on the full ASyMOB dataset. Prices vary mostly due to the vendor and the cost of tool use.

In terms of LLM queries, ASyMOB's testing cost is kept in check by using a single LLM call per problem, unlike many prior works, which assess models using protocols that involve multiple calls per problem. For instance, Minerva (Lewkowycz et al., 2022) is evaluated on the MATH dataset (Hendrycks et al., 2021), containing 12.5K problems, using maj1@k majority voting with k values up to 256, resulting in a minimum of 3.2 million LLM calls. The ASyMOB assessment approach is roughly 90X more cost-efficient due to our use of pass@1. The inherent LLM randomness is accounted for by evaluating success across the large number of questions within each category.

# C. Data Analysis

## C.1. Seed Performance

Figure 5 presents a color-coded summary of model performance across seed questions, averaged over all perturbations originating from that seed. The figure indicates that while certain seed questions are systematically more challenging than others - visible as lighter vertical bands, where most or all models struggle with variants derived from the same seed - strong models generally maintain a consistent performance advantage across nearly all seed questions. This advantage persists across different perturbation categories and mathematical domains.

In a small number of cases, specific seed questions and their variants appear to be disproportionately difficult for particular models, resulting in anomalously weak performance relative to the difficulty level experienced by other models on the same seed. Conversely, variations originating from some seed questions are solved almost perfectly by certain models - largely independent of the applied perturbations - even by models that do not rank among the top-performing setups. A detailed investigation of the underlying causes and structural patterns behind these anomalies is left for future work.

Figure 8 presents each model/setup's success on each seed question - showing a mix of easier and harder challenges.

## C.2. Seed Topics

To check that ASyMOB's aggregate conclusions are not driven by a single mathematical topic, we grouped all questions by the five mathematical categories: Integrals, Differential Equations, Series, Limits, and Hypergeometrics (Table 5).

*Table 5.* Number of seed questions and generated ASyMOB variants per mathematical topic.

| Category | Seed Questions | Total Questions |
|---|---|---|
| Integrals | 30 | 10694 |
| Differential Equations | 23 | 10157 |
| Series | 22 | 6994 |
| Limits | 15 | 4505 |
| Hypergeometrics | 10 | 3018 |
| Total | 100 | 35368 |

Table 6 summarizes model outputs after partitioning questions by mathematical topic. Conceptually, it asks whether each model/setup is uniformly strong across ASyMOB or whether some mathematical topics are substantially easier or harder for the model. The notable deviations from overall performance are GPT-4o (code)'s strong results on Hypergeometrics (49.6%, +14.3 points above its 35.3% total) and Llama-4-Scout-17B's weak results on the same topic (23.1%, -11.2 points below its 34.3% total), indicating that model families may differ in which symbolic structures they are able to handle robustly.

This analysis also reveals whether certain topics in the dataset are easier or harder than others. Averaged across model setups, Limits are the easiest topic (59.5%) - most models outperform their overall success rate by 10% or more, while Integrals are the hardest (42.6%).

## C.3. Question Independence and Variance

Generated variants derived from the same seed are not fully independent: they share the same original mathematical structure and topic. However, a purely seed-level analysis would also be too conservative since the perturbations produce distinct instances, introducing some level of statistical independence. We therefore use an intermediate diagnostic: first control for the largest predictable effects, and then ask how much seed-level structure remains in the residuals.

The two main predictable effects are question difficulty and model strength. We control for question difficulty because some generated questions are simply harder for nearly all models - if many models fail on variants from the same difficult seed, this should not automatically be interpreted as evidence that the generated variants add no new information. We estimate each question's difficulty from the performance of the other model/setups on that same question. We also control for model/setup strength: following the same logic, if a model is weak and fails on many question variants this shouldn't be interpreted as strong dependence between variants. After these controls, we compute residuals, defined as observed

*Table 6.* Topic-level model performance on ASyMOB in success percentages. The Total column is performance on the full dataset.

| Model | Integrals | Differential Equations | Series | Limits | Hypergeom. | Total |
|---|---|---|---|---|---|---|
| o4-mini (code) | 67.8 | 72.3 | 83.0 | 92.1 | 78.7 | 76.1 |
| o4-mini (no code) | 70.2 | 75.2 | 81.6 | 94.1 | 83.1 | 78.1 |
| Gemini-2.5 Flash (code) | 62.8 | 69.1 | 72.1 | 79.4 | 73.4 | 69.5 |
| Gemini-2.5 Flash (no code) | 69.8 | 79.9 | 79.4 | 92.2 | 81.8 | 78.5 |
| Gemini-2.0 Flash (code) | 52.2 | 61.2 | 53.8 | 75.6 | 52.1 | 58.1 |
| Gemini-2.0 Flash (no code) | 46.5 | 61.4 | 48.6 | 75.1 | 47.1 | 54.9 |
| GPT-4.1 (code) | 41.2 | 41.4 | 47.0 | 62.3 | 54.7 | 46.2 |
| GPT-4.1 (no code) | 43.0 | 47.8 | 47.5 | 66.4 | 50.7 | 48.9 |
| GPT-4o (code) | 35.3 | 28.6 | 30.4 | 48.7 | 49.6 | 35.3 |
| GPT-4o (no code) | 17.4 | 16.2 | 14.2 | 21.2 | 16.3 | 16.8 |
| DeepSeek-R1 | 72.3 | 73.7 | 85.0 | 91.5 | 79.5 | 78.3 |
| DeepSeek-Prover-V2-671B | 40.9 | 43.0 | 47.6 | 65.0 | 45.3 | 46.3 |
| DeepSeek-V3 | 39.6 | 42.5 | 45.1 | 65.9 | 46.1 | 45.4 |
| Qwen2.5-72B-Instruct | 27.5 | 27.9 | 27.6 | 34.9 | 23.6 | 28.2 |
| Llama-4-Scout-17B | 33.7 | 34.5 | 32.5 | 45.8 | 23.1 | 34.3 |
| Gemma-3n-e4b-it | 13.0 | 13.5 | 11.4 | 11.9 | 5.4 | 12.0 |
| GPT-4o-mini | 11.7 | 10.7 | 11.9 | 16.7 | 8.1 | 11.8 |
| Nemotron-Super-49B-v1 | 21.5 | 22.0 | 23.8 | 31.9 | 16.4 | 23.0 |
| Average across model setups | 42.6 | 45.6 | 46.8 | 59.5 | 46.4 | 46.8 |

correctness minus expected correctness, and aggregate these residuals by seed. If variants from the same seed still move together after accounting for question difficulty and model strength, the seed-level residual sums will be large.

Let $Y_{m,q} \in \{0,1\}$ denote whether model/setup $m$ solved generated question $q$, and let $s(q)$ be the original seed from which $q$ was generated. For each pair $(m,q)$, we estimate question difficulty from all other model/setups using a Jeffreys-smoothed leave-one-out rate (Jeffreys, 1961),

$$d_{m,q} = \frac{\sum_{m' \neq m} Y_{m',q} + 1/2}{|\{m' : m' \neq m\}| + 1}.$$

We then fit model/setup-specific intercept $a_m$ so that

$$\frac{1}{n_m} \sum_q \sigma(a_m + logit(d_{m,q})) = \frac{1}{n_m} \sum_q Y_{m,q},$$

where $\sigma$ is the logistic function. The expected correctness probability is therefore

$$\hat{p}_{m,q} = \sigma(a_m + logit(d_{m,q})),$$

and the residual is $e_{m,q} = Y_{m,q} - \hat{p}_{m,q}$.

For each setup, we estimate seed-clustered residual uncertainty by summing residuals within each seed:

$$\widehat{SE}_{seed}(m) = \frac{1}{n_m} \sqrt{\frac{S}{S-1} \sum_{s=1}^{S} \left( \sum_{q:s(q)=s} e_{m,q} \right)^2},$$

with $S = 100$ seeds. The factor $\frac{S}{S-1} = \frac{100}{99}$ is the finite-cluster correction; its numerical effect is small but avoids using the raw asymptotic cluster variance.

We convert this residual uncertainty into an IID-equivalent sample size by asking how many ideal homogeneous Bernoulli questions with the same observed success rate would produce the same standard error:

$$N_{eff,m} = \frac{\hat{p}_m(1 - \hat{p}_m)}{\widehat{SE}_{seed}(m)^2}.$$

This quantity should be interpreted as a comparison scale, not as a literal count of independent heterogeneous math questions.

The generated dataset contains 100 seed clusters and 35,368 generated variants. Cluster sizes range from 195 to 483 variants per seed, with mean 353.7 and median 299, so no single seed dominates the benchmark.

Treating all generated variants as fully IID gives unrealistically narrow 95% intervals of roughly 0.5%. After controlling for question difficulty and model/setup strength, the residual seed-clustered intervals average roughly 2.4%, corresponding to about 1440 IID-equivalent homogeneous Bernoulli questions. This supports the intermediate interpretation: ASyMOB is clearly structured by its 100 seeds, but the generated perturbations contain substantially more independent information than the 100 original seed questions alone.

As a sanity check, we also examined whether residual seed effects recur across perturbation families. They do: seeds that are unusually easy or hard under one generated perturbation family often remain partially easy or hard under others. Across model/setups, the median residual correlation is 0.78 for Symbolic–Numeric pairs and 0.73 for Numeric–Variance pairs, while correlations involving the unperturbed seed questions are lower, around 0.30–0.43 depending on the perturbation family. This suggests that seed identity still matters, but not in a way that collapses all generated variants back to the original seed outcome. Within the random numeric-variance subsets, split-half checks after conditioning on each seed's own mean success rate give dispersion values between 0.68 and 1.58, which is broadly consistent with substantial within-seed variation rather than perfect replication of a single seed-level result.

As an additional statistical test, Figure 9 illustrates the variance within each 100-question subset of variants 'Numeric-All-2-S' and 'Numeric-All-3-S' (per seed). It is important to note that while correct answers are unique (aside from presentation differences), incorrect answers can vary significantly, including instances where no answer is provided. Consequently, low consistency might result in lower variance for questions with a low success rate compared to those with a high success rate. Indeed, the average variance for all questions with success rate higher than 50% (for the specific model) is 0.11, whereas for questions below 50%, it is 0.07.

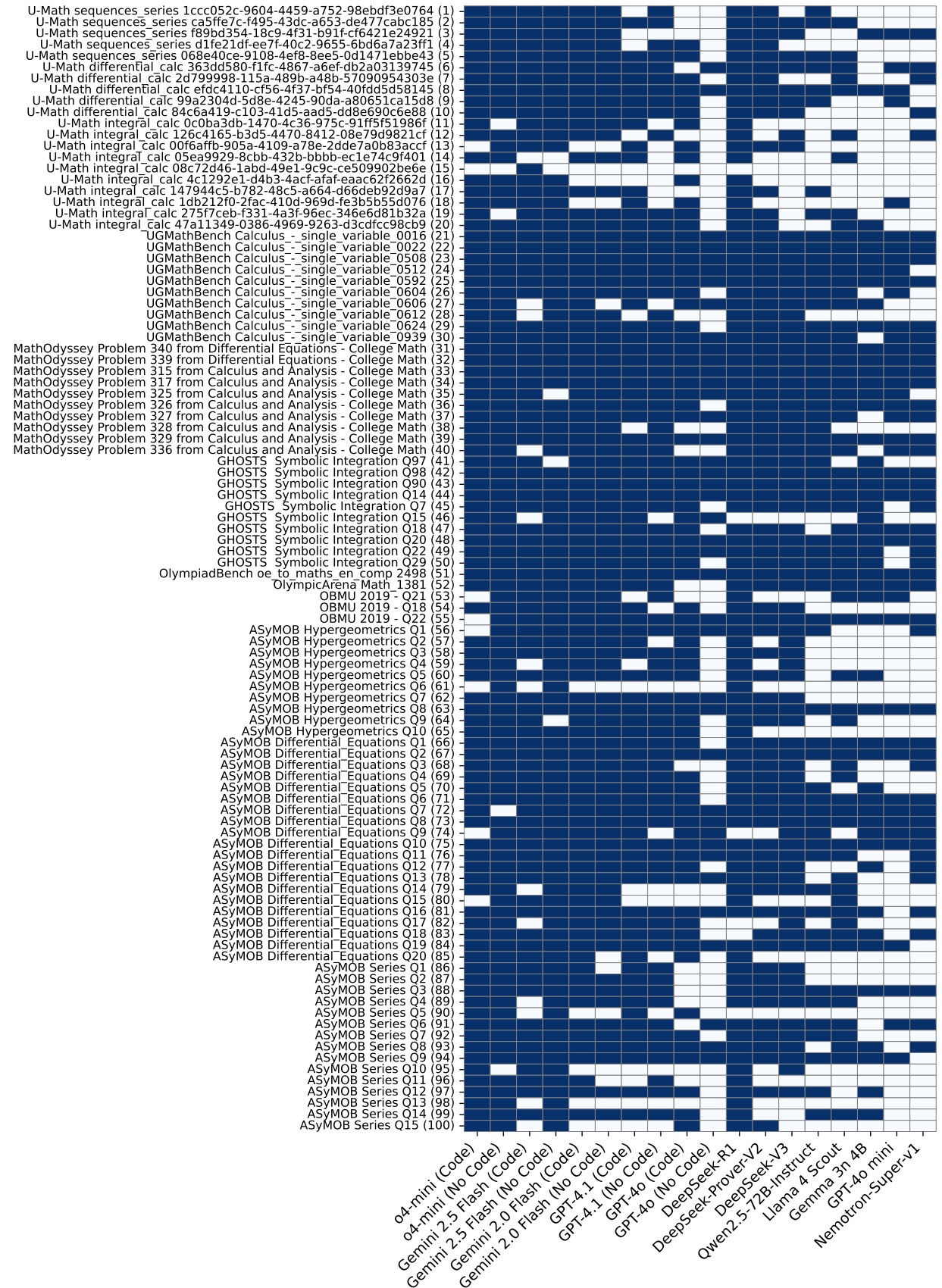

*Figure 8.* **Model success (blue) / failure (white) per seed question.** Seeds are marked by their source and index in the dataset. Note the difference in challenge level between seeds with different sources. 'ASyMOB' source indicates original questions that were created for the purpose of this work.

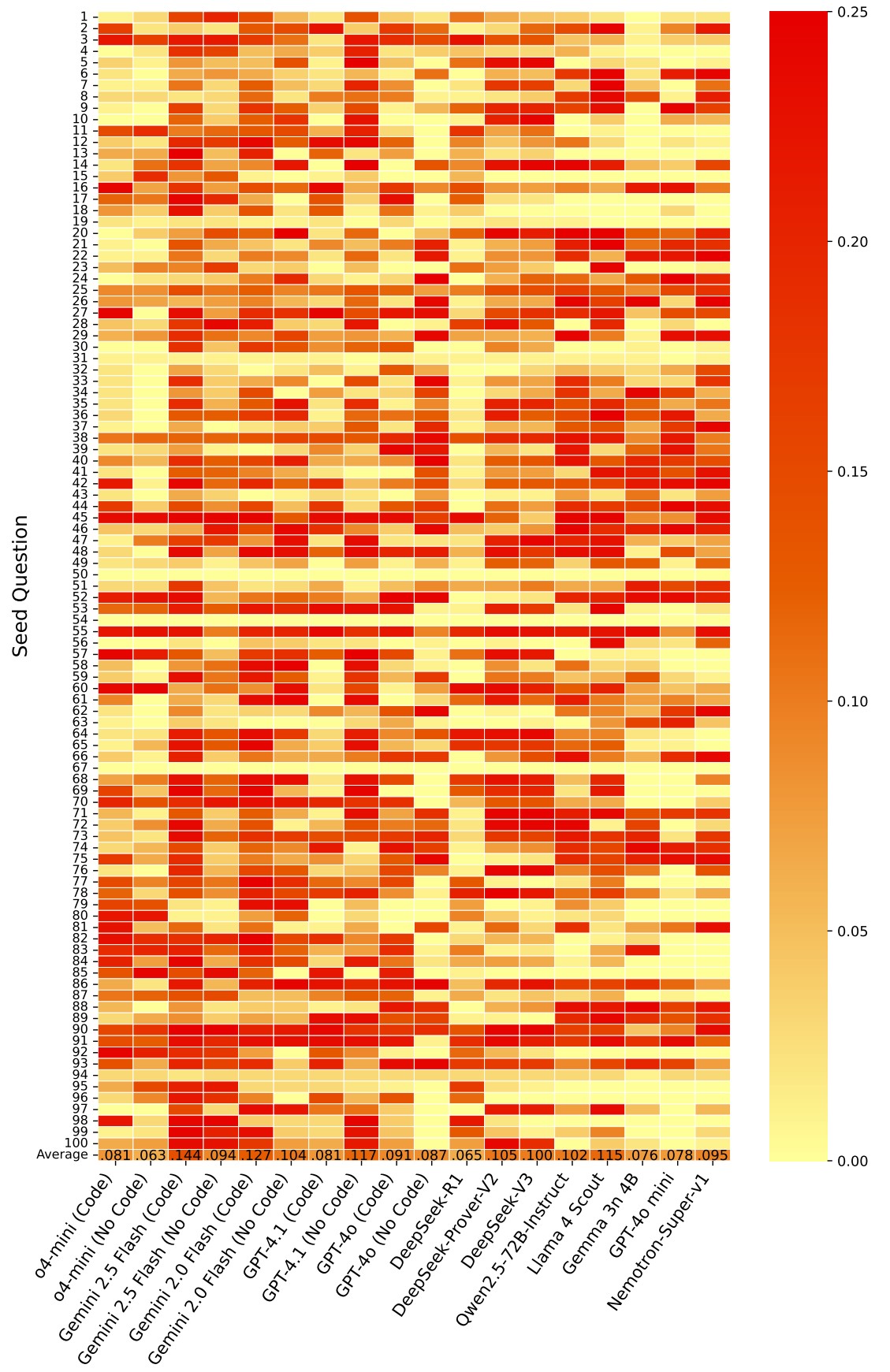

*Figure 9.* **Variance per model per seed question.** The bottom row shows the average variance of each model across all questions.

