# OpenReview forum: "ASyMOB: Algebraic Symbolic Mathematical Operations Benchmark"
_ICML.cc/2026/Conference — ICML 2026 regular_

### Official Review · Reviewer_amCJ · 2026-03-10

**Soundness:** 2
**Presentation:** 2
**Significance:** 2
**Originality:** 3
**Overall Recommendation:** 3
**Confidence:** 4

**Summary:**

This paper presents ASyMOB, a benchmark designed to evaluate symbolic mathematical manipulation by testing robustness under controlled perturbations. The benchmark focuses on symbolic reasoning tasks and includes a large set of instances generated from 100 seed problems. The authors propose a dual validation pipeline that combines symbolic and numeric checks, offering a more nuanced evaluation of model performance compared to standard exact-match metrics. The paper also explores the effectiveness of this benchmark for evaluating the robustness of LLMs and hybrid LLM+CAS systems in handling symbolic manipulation tasks.

**Compliance With Llm Reviewing Policy:**

Affirmed.

**Final Justification:**

My final recommendation remains 3 (Weak Reject). I continue to see the paper as having real benchmark-level originality, but after reading the rebuttal I do not think my main concerns were fully resolved. The rebuttal mainly offers clarifications and promised revisions rather than new evidence, so it does not change my core assessment on soundness and significance: the benchmark’s large size still comes from perturbing only 100 seed problems, the evaluation pipeline still leaves some reliability questions, and I still find several of the broader conclusions about understanding, memorization, and robustness stronger than the presented evidence supports. Overall, I view this as a worthwhile but still somewhat narrow benchmark contribution whose merits do not, in the current form, outweigh the remaining weaknesses, so I keep my original overall recommendation unchanged.

**Key Questions For Authors:**

1. While the benchmark contains 35,368 instances, these are derived from only 100 seed problems. Could the authors provide a more detailed statistical analysis at the seed level (e.g., seed-wise performance variability, confidence intervals), rather than treating the entire dataset as uniformly diverse?

2.The paper frequently interprets perturbation sensitivity as evidence of limited understanding. Could the authors clarify how they control for other potential causes of performance degradation, such as formatting issues, tokenization effects, or prompt mismatch?

3. The paper suggests that perturbations make the benchmark resistant to contamination. Could the authors provide a more detailed analysis of the types of contamination this approach guards against and its limitations?

**Limitations:**

No. The authors should provide a more detailed discussion on the potential biases introduced by the seed set, the reliability of the evaluation pipeline, and the risk of over-interpreting performance degradation as a reflection of "true understanding." Additionally, a broader consideration of potential societal impacts, such as the ethical implications of using LLMs for symbolic reasoning and the risks of reinforcing biases in AI, would improve the transparency and overall depth of the paper.

**Strengths And Weaknesses:**

Soundness
The empirical evidence does not fully support the paper’s stronger conclusions. The benchmark is large only in terms of generated instances because it still rests on 100 seed problems, which limits the strength of the statistical claims. The evaluation pipeline also depends partly on an LLM-based parsing fallback, yet the paper provides little evidence that this component is sufficiently reliable. As written, the claims about genuine understanding, memorization, and a possible robustness regime shift are overstated relative to the evidence.

Presentation
The paper is not sufficiently disciplined. It often moves from performance drops under perturbation to broad claims about reasoning deficits without clearly separating empirical findings from interpretation. The framing is also imprecise: the work is fundamentally a benchmark for symbolic manipulation, yet the discussion sometimes suggests broader conclusions about formal theorem proving than the setup justifies.

Significance
The paper addresses a real but relatively narrow evaluation gap. A benchmark for symbolic manipulation may only be useful for analyzing robustness in algebraic settings. The paper does not introduce a new capability, method, or evaluation paradigm with clear implications for the wider formal reasoning or theorem proving literature. As a result, the significance is moderate and should not be overstated.

Originality
The paper offers meaningful originality at the benchmark level. Its main contribution is not methodological innovation in symbolic reasoning, but the construction of a targeted evaluation setting for pure symbolic manipulation under controlled perturbations. That design distinguishes the work from more generic math benchmarks. However, the paper does not introduce a new solver, a new symbolic reasoning framework, or a fundamentally new evaluation principle. The originality is therefore real, but appropriately understood as benchmark novelty rather than broader algorithmic innovation.

---

> ### Author Rebuttal · Authors · 2026-03-30
>
> *“...depends partly on an LLM-based parsing fallback…little evidence that this component is sufficiently reliable.”*
>
> To clarify, the fallback is used only for parsing extracted LaTeX into SymPy, not for verification (unlike prior work). Cases failing translation are anyway excluded from analysis so our conclusions hold even under worst-case assumptions. We will clarify this in Section 2.2.
>
> *“...claims about genuine understanding…are overstated relative to the evidence.”*
>
> We agree "genuine understanding" is a philosophically contested term and will replace it with quantitative results.
> Our novelty lies in quantifying necessary conditions for symbolic competence, e.g., maintaining correctness under perturbations that preserve underlying mathematics. Thus, more than previous works, ASyMOB provides measurable hints to a certain level of “understanding” (or lack thereof).
>
> *“...broad claims…without clearly separating empirical findings from interpretation”*
>
> We accept this criticism. The final text will clearly separate empirical findings from interpretation.
>
> *”…broader conclusions about formal theorem proving than the setup justifies.”*
>
> To which part the reviewer is referring? The claims in Section 3 regarding proof proficiency come from the cited papers.
>
> *“...a real but relatively narrow evaluation gap…only useful for analyzing robustness in algebraic settings.”*
>
> This focus is deliberate. Existing benchmarks mix text interpretation, arithmetic, proof, and symbolic manipulation; ASyMOB isolates a single core capability to quantify robustness.
> We believe future benchmarks will follow this specialization trend - preferring focused quantitative results over broader qualitative ones.
> Moreover, algebraic manipulation is a cornerstone of STEM, with broader importance than many other areas of pure math.
>
> *“...does not introduce a new solver, a new symbolic reasoning framework, or a fundamentally new evaluation principle.”*
> *“The paper does not introduce a new capability, method, or evaluation paradigm…”*
>
> ASyMOB introduces a new evaluation paradigm (not a solver or reasoning framework): assessing symbolic competence via stability under controlled perturbations, rather than accuracy on static items. This enables quantitative analysis of degradation relative to seed performance - as the reviewer said: “targeted evaluation setting”. We will emphasize this focus in the manuscript.
>
> *“...100 seed problems, which limits the strength…”*
> *“...provide…statistical analysis at the seed level…rather than treating the entire dataset as uniformly diverse?”*
>
> We do not treat the dataset as uniformly diverse. All degradation claims are measured relative to seed performance, which normalizes for inherent seed difficulty. Furthermore, Appendix C explicitly details seed-wise heterogeneity.
> However, for deeper analysis, we will add: (1) Seed-wise confidence intervals, complementing Appendix C (2) Topic-level aggregation across the five categories to show consistent trends.
>
> *“The paper frequently interprets perturbation sensitivity as evidence of limited understanding. Could the authors clarify how they control for other potential causes of performance degradation, such as formatting issues, tokenization effects, or prompt mismatch?”*
>
> All variants use identical prompt and formatting structures (Appendix B) - removing prompt mismatch and formatting issues by construction.
> We view tokenization sensitivity (e.g. expression swell) as part of the measured phenomenon, not as a confounding variable. If a model fails on mathematically identical logic due to token-level brittleness, that is precisely the fragility ASyMOB is designed to expose. We will clarify this stance in the text.
>
> *“The paper suggests that perturbations make the benchmark resistant to contamination. Could the authors provide a more detailed analysis of the types of contamination this approach guards against and its limitations?”*
>
> ASyMOB is designed to mitigate risk of memorization of fixed Q-A pairs from public datasets, by regenerating the questions at negligible cost.
> Our Section 4 stress test (using seed Q-A as in-context exemplars) shows that even direct exposure does not resolve the difficulty of complex variants.
> We will further clarify the distinction between data contamination - which we mitigate - and methodological contamination (e.g., training on the perturbation framework itself) - which we don’t.
>
> *“...broader consideration of potential societal impacts, such as the ethical implications of using LLMs for symbolic reasoning and the risks of reinforcing biases in AI…”*
>
> We thank the reviewer and will expand the Impact Statement to include broader societal considerations.
> Benefits include democratizing scientific discovery through robust symbolic tools. A primary risk is automation bias - users over-trusting plausible but brittle outputs. ASyMOB directly addresses this by improving trust calibration and encouraging rigorous verification.

---

> > ### Author Rebuttal · Reviewer_amCJ · 2026-04-04
> >
> > Some concerns were clarified, but the remaining ones still concern the core evaluation methodology. In particular, I still think the paper needs stronger seed-level statistical support given that the benchmark is derived from 100 seed problems, and the discussion of the LLM-based parsing fallback would benefit from direct reliability or sensitivity analysis. These issues seem to require additional analysis and corresponding revision, rather than clarification alone.

---

### Official Review · Reviewer_HkXJ · 2026-03-12

**Soundness:** 3
**Presentation:** 3
**Significance:** 2
**Originality:** 2
**Overall Recommendation:** 4
**Confidence:** 3

**Summary:**

The paper presents a benchmark for algebraic symbolic math questions. The dataset is dynamically generated from a set of seed questions. In their experiments, the authors use 100 seed questions covering integrals, differential equations, series, limits, and hypergeometric functions. Then, they manually introduce 2 to 5 symbolic parameters into each seed questions after which they generate additional variants using algorithmic transformations. Specifically, they create variants labeled "Symbolic-N" where the previously introduced symbolic parameters are replaced with numeric parameters (consequently, N is between 2 and 5). These numeric variants consider positive integers of fixed digit length (varying from 0 to 10 digits). Additionally, the authors evaluate the impact of equivalent-form perturbations by inserting one or more expressions that are mathematically equal to 1. Specifically, those identity types include trigonometric, hyperbolic, logarithmic, complex exponentials, and series. In total this yields 35,368 questions.

The authors then evaluate a set of open-weight and black-box models on those questions. They also separate when models are given access to code execution tools (for black-box models they instruct the models to either use or not use them). Evaluation is done using pass@1 and a sympy regex parser. The authors have a number of findings, such as performance degrading with the number of perturbations introduced, SOTA models being robust to symbolic perturbations and being less reliant on code execution (although they can't verify that they did not use them). Tool use does, however, improve performance on weaker models.

**Compliance With Llm Reviewing Policy:**

Affirmed.

**Final Justification:**

The benchmark introduced is useful to evaluate the effect of controlled perturbations on model performance. The findings broadly suggest that models without tool-use struggle with mathematical equations. This is, however, similar to findings many prior papers reported.

In principle, this approach could be applied to other benchmarks. But this requires human labor in its current form, since the authors manually perturbed each seed question with 2-5 symbolic parameters (L151). It remains unclear if LLMs could be used to automate this step (i.e. if they are able to identify meaningful positions at which one might introduce those parameters).

**Key Questions For Authors:**

NA

**Limitations:**

Yes

**Strengths And Weaknesses:**

**Strengths**
- Dynamically generating a benchmark from a set of seed questions is a useful way of circumventing memorisation.
- The experiments and analyses appear well-executed and presented.
- The paper is generally well-written and easy to follow.

**Weaknesses**
- The fact that perturbations significantly challenge LLMs symbolic math skills is interpreted as reliance on pattern memorisation and lack of "true understanding" (see Sec. 4). However, Table 1 suggests that each perturbation just makes the expression longer, thus harder. So an alternative reading is that model performance degrades with the length of the algebraic equation, which seems quite plausible.
- The authors argue that regenerating the benchmark for each evaluation would prevent memorisation. However, the dataset is still generated from the same 100 seed questions. While this meaningfully tests some degree of generalisation, it doesn't prevent all forms of memorisation (e.g. memorising that some equivalent-form perturbations can be ignored).
- Some of the design choices seem arbitrary: While numeric variants allow replacing any number of parameters, the equivalent-form perturbations perturb either all symbols or only one. Similarly, classification into "hard" and "easy" isn't well motivated (only retroactively by a drop in model performance).
- Some claims in the discussion around CAS seem unsubstantiated. For example, the authors argue that their results show that "once an LLM learns when and where to use tools, it can mitigate substantial pitfalls by using code execution as a form of grounding". While the results show that models with code execution generally outperform their counterparts without code execution (albeit mostly for weaker models) it doesn't show *why* this is the case.

---

> ### Author Rebuttal · Authors · 2026-03-30
>
> - *"Dynamically generating a benchmark from a set of seed questions is a useful way of circumventing memorisation."*
> - *"The experiments and analyses appear well-executed and presented."*
> - *"The paper is generally well-written and easy to follow."*
>
> We thank the reviewer for their positive feedback.
>
> *“...an alternative reading is that model performance degrades with the length of the algebraic equation”*
>
> While the expression length might play a role in model performance (as we discussed in Section 3), we respectfully note that the relationship between length and performance is more nuanced. In our numeric‑0 variants (substituting symbols by 1), adding just two characters (“1*” or “^1”) to certain expressions of 49-70 characters (3-4% increase in length) causes substantial degradation in performance (50-100% relative to the seed performance). In another question, adding more characters (32) to a shorter expression (48) causes no degradation in performance.
>
> Another example of the non-proportionality between length and performance is seen when  comparing symbolic and numeric‑1 perturbations (single‑digit constants) on the same question - extending its length by the same amount. This test isolates the effect of perturbation type from length. We observe a distinct performance gap - symbolic perturbations are harder - which confirms that models struggle with the mathematical substitutions, and not merely with token count.
>
> We will add a paragraph to Section 4 to clarify the nuanced role of token length in symbolic math performance degradation.
>
> *“The authors argue that regenerating the benchmark…would prevent memorisation. …While this meaningfully tests some degree of generalisation, it doesn't prevent all forms of memorisation (e.g. memorising that some equivalent-form perturbations can be ignored).”*
>
> We agree. In fact, no benchmark can fully rule out memorization. Our claim is narrower: ASyMOB’s dynamic generation reduces this risk relative to static benchmarks. As the reviewer notes, it “meaningfully tests some degree of generalisation”. This is exactly our goal. We strongly believe that *quantitative* tests of certain “degree(s) of generalisation” would provide valuable benchmarks, more than the existing less focused tests.
>
> Section 4’s contamination stress test supports this point. Even when the seed question and answer are given as an in-context exemplar, the more complex variants remain challenging.
>
> Regarding the concern that a model might learn to ignore certain equivalent-form perturbations, note that all equivalence perturbations were correctly resolved to 1 when presented in isolation (Appendix A.1). In other words, models can already handle these forms individually; the difficulty arises when the same simplifications must be applied within a more complex, multi-step expression.
>
> *“...the equivalent-form perturbations perturb either all symbols or only one. Similarly, classification into "hard" and "easy" isn't well motivated”*
>
> Our choice to perturb either "one" or "all" symbols for equivalent-form perturbations was designed to bound the complexity of the problem space. By testing the minimum and maximum extremes, we establish the boundaries of the perturbation's effect without causing a combinatorial explosion in evaluation costs or disproportionately skewing the dataset balance.
>
> Regarding the "hard" and "easy" classification, we will revise the manuscript to clarify that this was motivated a priori by the intrinsic mathematical complexity of the perturbations (the number of nested operations in the expression), which was then retroactively validated by the drop in model performance.
> Further exploration of the relation between symbolic complexity metrics and LLM performance on these challenges is left for future work.
>
> *“...the authors argue that their results show that 'once an LLM learns when and where to use tools, it can mitigate substantial pitfalls by using code execution as a form of grounding'.”*
>
> The quoted sentence was intended as a hypothesis rather than a proven statement, and we will rephrase it to better reflect the evidence we have.
> However, it is motivated by our measured data. Just as previous works demonstrate that LLMs struggle with direct arithmetic calculations but succeed when writing code to perform those same calculations, we observe a similar behavioral effect here for symbolic math.
>
> *“While the results show that models with code execution generally outperform their counterparts without code execution…it doesn't show why this is the case.”*
>
> Being a benchmark project, ASyMOB provides quantitative information and does not address the “why”, which will require complete access to the neural networks. Still, the ASyMOB quantitative approach provides unique information for deeper analysis.
> e.g. the ‘statistical’ subsets (‘Numeric-All-2/3-S’) allow for a quantification of the change in model performance *consistency* when allowing code-use (Fig 9).

---

> > ### Author Rebuttal · Reviewer_HkXJ · 2026-04-02
> >
> > Thanks for the thorough response and clarifications. I will increase my score to 4.

---

### Decision · Program_Chairs · 2026-04-30

**Decision:**

Accept (regular)

**Comment:**

This paper presents ASyMOB, a benchmark of 35,368 validated symbolic math problems generated from 100 seed problems via systematic perturbations (symbolic, numeric, and equivalence-preserving transformations). The benchmark isolates pure symbolic manipulation from text comprehension, enabling fine-grained measurement of LLM robustness across controlled difficulty levels. The primary takeaways include that most models degrade substantially under even minor perturbations, frontier models exhibit a possible regime shift in robustness, and tool use stabilizes weaker but not stronger models.
Reviewer HkXJ found the benchmark useful and well-executed, noting that dynamic generation meaningfully circumvents memorization. Their concern that performance degradation may reflect expression length rather than reasoning fragility was addressed in the rebuttal with evidence that small length changes (3-4%) cause disproportionate degradation, and that symbolic vs. numeric perturbations of equal length produce distinct performance gaps. This reviewer marked concerns as fully resolved.
Reviewer amCJ raised concerns about the 100-seed foundation limiting statistical claims, the reliability of the LLM-based parsing fallback, and overclaiming about "genuine understanding." The authors clarified that the parsing fallback is used only for LaTeX-to-SymPy translation (not verification), with failed cases excluded from analysis. They also committed to replacing overclaiming language with quantitative framing. However, this reviewer maintained a weak reject, viewing the contribution as too narrow.
This paper received only two reviews despite efforts to assign a third, so I have read the paper in detail to form an independent assessment. I find the benchmark methodology to be well-designed and the evaluation thorough. The perturbation-based approach introduces a genuinely useful evaluation paradigm for symbolic math, distinct from prior work (GSM-Symbolic, MATH-Perturb) that focuses on word problems or school-level math. The dual validation pipeline avoids LLM-as-judge pitfalls, and the contamination stress test (Section 4) is a thoughtful addition. The 100-seed concern is legitimate but not disqualifying: GSM-Symbolic, the seminal perturbation-based math benchmark, also used 100 seeds. The conceptual contribution of controlled perturbation-based evaluation for university-level symbolic math is the primary novelty, and the current scale is sufficient to support the paper's main conclusions. I do agree with Reviewer amCJ that some interpretive claims should be toned down to match the evidence, and I expect the authors to follow through on this in the camera-ready.